# Cytoskeletal rearrangement precedes nucleolar remodeling during adipogenesis
Evdokiia Potolitsyna[1,3], Sarah Hazell Pickering [1,4], Aurélie Bellanger [1,4], Thomas Germier[1], Philippe Collas[1,2] & Nolwenn Briand [1] ✉

Differentiation of adipose progenitor cells into mature adipocytes entails a dramatic reorganization of the cellular architecture to accommodate lipid storage into cytoplasmic lipid droplets. Lipid droplets occupy most of the adipocyte volume, compressing the nucleus beneath the plasma membrane. How this cellular remodeling affects sub-nuclear structure, including size and number of nucleoli, remains unclear. We describe the morphological remodeling of the nucleus and the nucleolus during in vitro adipogenic differentiation of primary human adipose stem cells. We find that cell cycle arrest elicits a remodeling of nucleolar structure which correlates with a decrease in protein synthesis. Strikingly, triggering cytoskeletal rearrangements mimics the nucleolar remodeling observed during adipogenesis. Our results point to nucleolar remodeling as an active, mechano-regulated mechanism during adipogenic differentiation and demonstrate a key role of the actin cytoskeleton in defining nuclear and nucleolar architecture in differentiating human adipose stem cells.

The morphological transition from a fibroblast-like adipose stem cell (ASC) into a lipid-filled mature adipocyte involves a significant reorganization of cellular architecture. Mature adipocytes display a lineage-specific cytoskeleton with a diffuse cortical actin network at the cell periphery and vimentin surrounding cytoplasmic lipid droplets[1–4]. Actin cytoskeleton remodeling is thus critical for adipogenic commitment and differentiation[5–8] and impaired cytoskeletal remodeling during adipogenesis results in differentiation defects[9,10].

The cytoskeleton is physically connected to the nuclear lamina, a meshwork of polymerized A- and B-type lamins, which interfaces the nuclear envelope and chromatin, via the Linker of Nucleoskeleton and Cytoskeleton (LINC) complex[11,12]. Hence, cytoskeletal remodeling during adipogenesis directly impacts nucleus size, morphology and positioning. In vivo, white adipocytes harbor an elliptic nucleus, pushed to the cell periphery by a single lipid droplet. In vitro, adipogenesis is accompanied by a decrease in nuclear volume, which has been proposed to result from a decrease in cell tension upon depolymerization of the actin network[13]. Accordingly, cell stiffness of human preadipocytes decreases from adipogenesis onset, and upon lipid accumulation[14].

While ASC differentiation has been extensively studied at the level of gene expression, epigenetic regulation and genome organization[15–18], the impact of nuclear remodeling on nuclear substructures has not been described. The largest nuclear subcompartment is the nucleolus, which structurally consists of a fibrillar center (FC), a dense fibrillar component (DFC), and a granular component (GC)[19,20]. Additional substructures have been recently described, such as the nucleolar rim on the periphery of the GC[21], and the "periphery of dense fibrillar component" between the DFC and the GC[22]. The nucleolus governs ribosome biogenesis and de novo protein synthesis. Recent reports suggest that the nucleolus is a mechanosensitive organelle responsive to changes in extracellular matrix stiffness and intracellular tension[23–25]. Accordingly, defective cytoskeletal remodeling during adipogenesis is accompanied by a deregulation of translation[9,10]. However, whether nucleolar function during adipogenesis is affected by nuclear remodeling remains unknown.

Here, we provide a detailed characterization of nucleolus remodeling during adipogenic differentiation of human ASCs. We find that structural remodeling of nucleoli in differentiating cells correlates with reduced rates of protein synthesis. Nucleolus remodeling precedes the change in nuclear size

[1]Department of Molecular Medicine, Institute of Basic Medical Sciences, Faculty of Medicine, University of Oslo, BlindernPO Box 1112, 0317 Oslo, Norway. [2]Department of Immunology and Transfusion Medicine, Oslo University Hospital, 0424 Oslo, Norway. [3]Present address: Department of Molecular and Cellular Biology, Baylor College of Medicine, Houston, TX 77030, USA. [4]These authors contributed equally: Sarah Hazell Pickering, Aurélie Bellanger. ✉e-mail: nolwenn.briand@medisin.uio.no

and is triggered by cytoskeleton rearrangement, pointing to an active, mechano-regulated, mechanism.

## Results

### Adipogenesis results in profound nuclear and nucleolar remodeling

Adipogenesis requires a remodeling of cellular shape from a fibroblast-like progenitor cell into a round, lipid-loaded mature adipocyte (Supplementary Fig. 1a, b). In contrast to proliferating ASCs (henceforth, Pro), confluent, growth-arrested undifferentiated ASCs (day 0 timepoint; D0) display a highly polymerized and well-organized network of actin stress fibers (Fig. 1a-d; Supplementary Fig. 1c). Exposure of D0 cells to the adipogenic differentiation cocktail activates a transcriptional cascade converging on the expression of the master regulators *PPARG* and *CEBPA*, which together promote and maintain the mature adipocyte phenotype (Supplementary Fig. 1c, d). Induction of adipogenesis triggers a marked remodeling of the cytoskeleton detected as early as D3, while only a diffuse and weakly labeled actin network remains by our differentiation endpoint on D15, in both donors examined (Fig. 1a–d; Supplementary Fig. 2a, b). Cytoskeleton remodeling is paralleled by a remodeling of the nuclear lamina, with D0 ASCs displaying the highest lamin A levels, consistent with an increase in intracellular tension[26,27] (Fig. 1a, e; Supplementary Fig. 2a, c, d). Further, lipid accumulation from D3 to D15 is accompanied by a significant (2-fold) increase in Lamin A levels, and decreased nuclear volumes attesting of alterations in intracellular tension also in late differentiation (Fig. 1a, e, f; Supplementary Fig. 2a, c, d, e). Indeed, nuclear volume and elongation show respectively a negative and positive correlation with lipid droplet size at D15 (Fig. 1f–i), indicating that the mechanical load exerted by the lipid droplet is a major driver for nuclear deformation in mature adipocytes.

We next assessed whether these changes in structural components of the cytoskeleton and the nuclear lamina would impact nucleolar architecture during adipogenesis. We used nucleolin as a GC marker, as protein levels of nucleolin do not vary significantly during differentiation (Supplementary Fig. 3a–c). Nucleolin immunostaining reveals a transition in nucleolar morphology, characterized by a change in nucleolus structure from D0 onwards, and the formation of a single nucleolus in D15 adipocytes (Fig. 2a; Supplementary Fig. 4a). We also note that after an initial drop upon growth arrest (D0), the total nucleolar volume steadily increases while the number of nucleoli decreases during differentiation (Fig. 2a–c; Supplementary Fig. 4a–c).

To assess changes in nucleolar volumes in relation with variations of nuclear size, we calculated the nucleolus-to-nucleus volume ratio (No/Nu), thereby normalizing for the changes in nuclear size across differentiation time-points. While the No/Nu ratio significantly decreases at D0, adipogenesis is associated with an increase in No/Nu ratio with single nucleoli occupying on average 9%, and up to 20%, of the nucleus volume in differentiated D15 adipocytes (Fig. 2d; Supplementary Fig. 4d). Importantly, this parallel reduction in the number of nucleoli and increase in No/Nu ratio is not observed when ASCs are induced to differentiate towards the osteogenic lineage (Supplementary Fig. 5a–c). This suggests that the nucleolar remodeling observed during adipogenic differentiation is lineage-specific. Altogether, our results indicate that the nucleus and nucleoli are both dramatically remodeled during adipogenesis, albeit with distinct kinetics.

### Nucleolar structure is remodeled in growth-arrested undifferentiated ASCs

We further assessed if the remodeling of nucleolar structure observed on D0 and onwards might affect the organization of nucleolar subcompartments. Proliferating and differentiating ASCs were immunostained with the GC marker nucleolin and UBTF, a marker of active FCs[28,29] (Supplementary Fig. 3a, d). We find that nucleoli of proliferating ASCs display tubular-like morphologies with interwoven Nucleolin and UBTF labeling (Fig. 3a, b). In

contrast, in D0 ASCs and during adipogenesis, the nucleolin signal is significantly enriched at the nucleolar periphery, and only partially overlaps with UBTF staining (Fig. 3a, b; Supplementary Fig. 6a). Such redistribution is also observed with Nucleophosmin (NPM1) and RNA POL1 (RPA194 subunit) co-immunostaining, other markers of the GC and FC respectively, arguing for global reorganization of nucleolar structure (Fig. 3c, d). Of note, redistribution of Nucleolin and UBTF does not correlate with a change in expression levels assessed by Western blotting (Supplementary Fig. 3a, c, d). In contrast, protein levels of RPA194 decrease together with the structural reorganization of the nucleolus (Supplementary Fig. 3a, e).

We further used super-resolution radial fluctuation (SRRF)-stream microscopy to visualize nucleolar substructures during differentiation[30]. SRRF-stream analysis enhances the observed redistribution of the GC marker Nucleolin to the nucleolar periphery and allows to resolve UBTF-marked FCs (Fig. 3e). UBTF-bound ribosomal DNA (rDNA) repeats constitute transcriptionally competent units and are essential determinants of nucleolar organization[31]. We find that the number of UBTF-marked foci is significantly reduced in growth-arrested D0 ASCs, as well as at an early differentiation time-point, suggesting that nucleolar remodeling is linked to a decrease in cellular activity (Fig. 3f). Accordingly, induction of cell-cycle arrest by 24-h serum deprivation in proliferating ASCs promotes the enrichment of Nucleolin at the nucleolar periphery similar to that of D0 ASCs, together with a reduction of cell and nuclear size (Fasted; Fig. 3g; Supplementary Fig. 6b–d). However, serum deprivation also results in a pronounced redistribution of Nucleolin into the nucleoplasm (Fasted; Fig. 3g), a hallmark of nucleolar stress we do not observe during differentiation (see Fig. 1e)[32,33]. As anticipated from a cell cycle-arrest phenotype, these changes are reversed upon serum re-addition, while Nucleolin levels are unchanged (Refed; Fig. 3g–i). Importantly, actin cytoskeleton organization is not affected in fasted ASCs, and nucleolar remodeling does not correlate with changes in nuclear size, which remains reduced after refeeding (Supplementary Fig. 6d, e). We conclude that changes in nucleolar substructures occurring during adipogenic differentiation result from cell cycle arrest and are independent from cytoskeleton remodeling.

### Adipogenic differentiation results in blunted translation efficiency

Given the primary function of the nucleolus as a ribosome factory, we next examined whether nucleolar remodeling affects ribosome biogenesis or translational activity during adipogenesis. To gain insight into these highly coordinated processes, we first assessed transcriptional changes in the ribosome biogenesis (Gene ontology GO:0042254) and translation (GO:0006412) pathways (Fig. 4a; Supplementary Fig. 7a). We find that genes involved in ribosome biogenesis segregate into 3 clusters: while cluster 1 and 2 genes are respectively down- and up-regulated between D3 and D15, cluster 3 genes are downregulated from D0 onwards (Fig. 4a). Similarly, adipogenic differentiation results in both a down-regulation in the expression of translation initiation factor genes (clusters 4 and 5) and in an up-regulation of translation elongation factors' gene expression (clusters 6 and 7) (Fig. 4a). Intriguingly, a large number of ribosomal protein (RP) genes (e.g. *RPL10a*, *RPS26*) are upregulated in differentiated adipocytes (Fig. 4a, cluster 7), in contrast to the overall decrease in the expression of genes involved in ribosome biogenesis (Fig. 4a, cluster 1 and 3; Supplementary Fig. 7a)[34,35]. Such transcriptional remodeling appears further exacerbated when analyzing mature adipocytes isolated by flotation from a D15 cell pool (Supplementary Fig. 7b). Conversely, it is not observed during osteogenic differentiation (Supplementary Fig. 7c), implying an adipose-specific regulation rather than a common feature of terminally differentiated cells. Overall, our transcriptome analysis reveals a profound remodeling of ribosome biogenesis and translation pathways.

To assess the functional impact on protein synthesis, we used a surface sensing of translation (SUnSET) assay which measures the incorporation of puromycin into nascent peptides[36] (Fig. 4b, c; Supplementary Fig. 8a). We find that translation levels are the highest in proliferating cells, and drop significantly at D0, as expected for growth-arrested cells. However, translational activity does not resume upon adipogenic induction, and remains

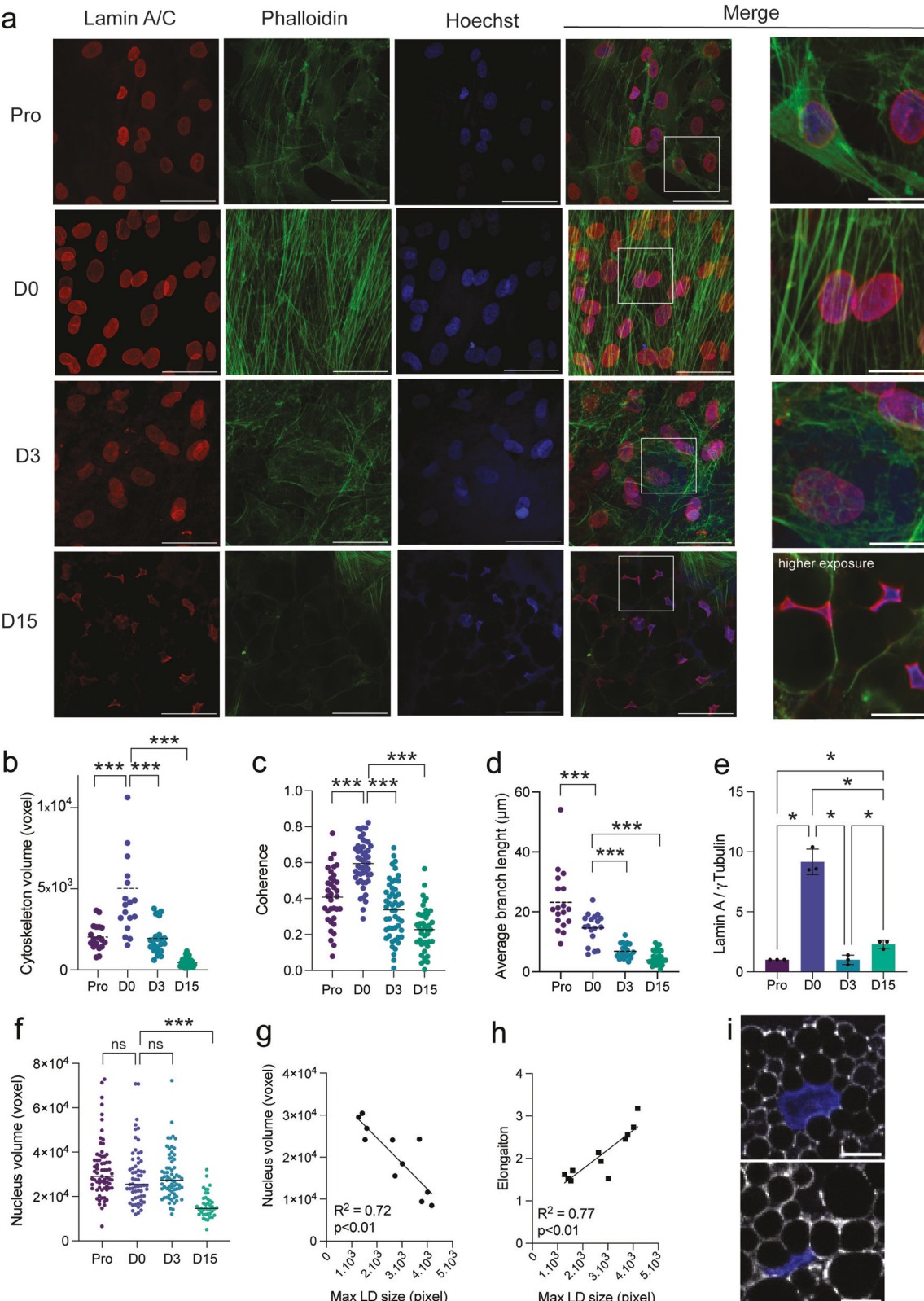

**Fig. 1 | Nuclear remodeling during adipogenesis. a** Immunofluorescence of lamin A/C, Phalloidin and DAPI stainings in differentiating ASCs scale bar: 10 μm). **b**, **c**, **d** Relative cytoskeleton volume, coherence, and average branch length measured from phalloidin signal (***$p$ < 0.0001 vs D0, two-way ANOVA with Tukey's multiple comparison test; n ≥ 5 fields of 2500 μm$^2$). **e** Lamin A signals, normalized to γTubulin, quantified from Western blots (*$p$ < 0.05, one-way ANOVA with Holm-Šídák's multiple comparisons; $n$ = 3 experiments). Data are presented as mean ± SD.

**f** Nuclear volumes (voxel) measured from DAPI signal (***$p$ < 0.001 vs D0, two-way ANOVA with Tukey's multiple comparison; n ≥ 30 cells per time-point from 3 experiments). **g**, **h** Scatterplots of nucleus volume (voxel) or nuclear elongation $vs$ maximal lipid droplet (LD) size (pixel), fit with linear regression. **i** Representative immunofluorescence images of Perilipin1 (PLIN1) and DAPI staining in D15 adipocytes (scale bar: 10 μm).

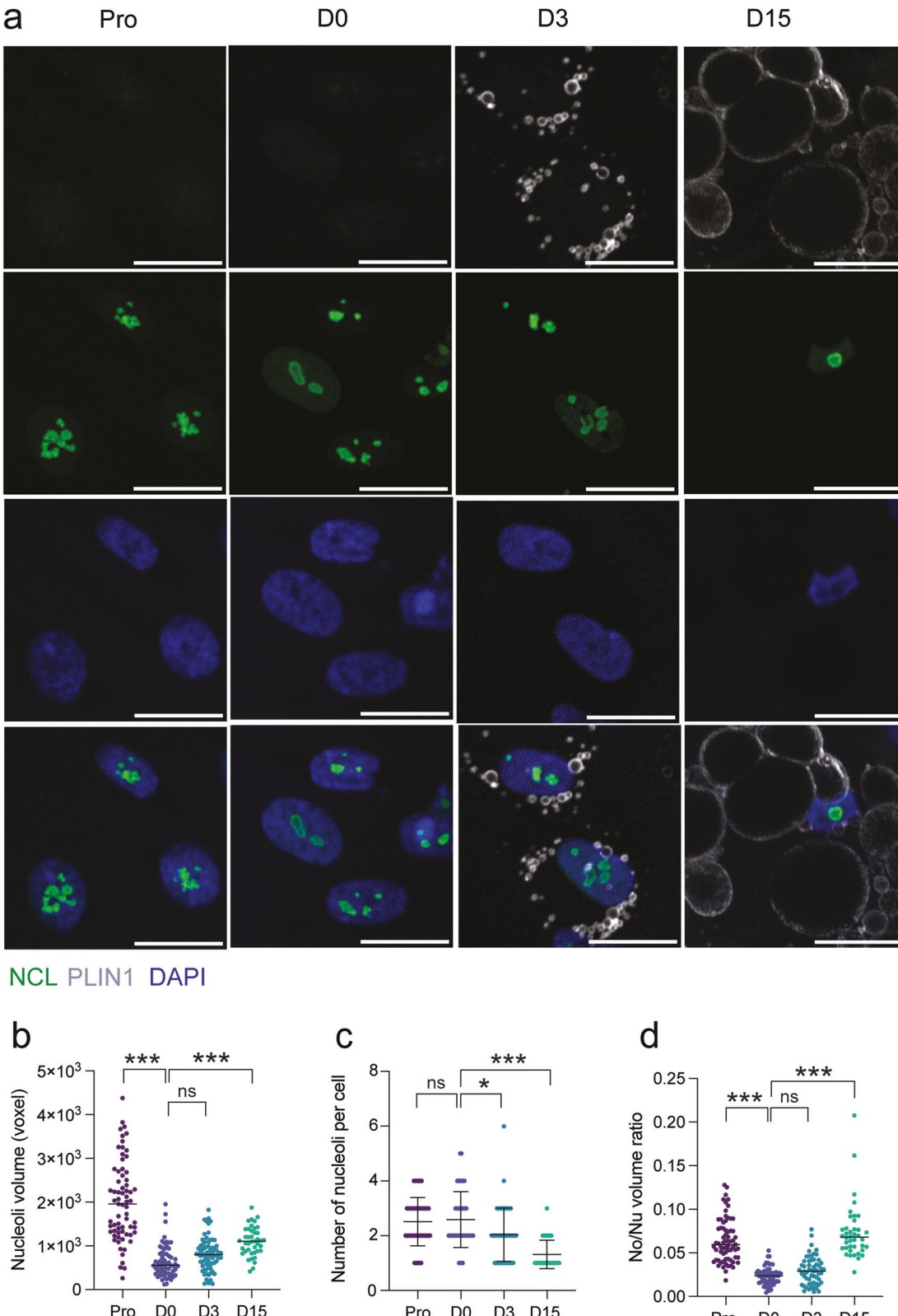

NCL PLIN1 DAPI

**Fig. 2 | Nucleolar remodeling during adipogenesis. a** Immunofluorescence of Nucleolin (NCL), Perilipin1 (PLIN1) and DAPI staining in differentiating ASCs (scale bar: 10 µm). **b** Scatter plot of nucleolar volume measured from Nucleolin immunostaining (***$p < 0.0001$ vs D0, two-way ANOVA with Tukey's multiple comparison; n ≥ 60 cells per time-point from 3 experiments). **c** Scatter plot of the number of nucleoli per cell (*$p < 0.05$, ***$p < 0.0001$ vs D0, two-way ANOVA with Tukey's multiple comparison test; n ≥ 60 cells per condition from 3 experiments). Data are presented as mean ± SD. **d** Scatter plot of nucleolus-to-nucleus volume (No/Nu) ratio (***$p < 0.0001$ vs D0, two-way ANOVA with Tukey's multiple comparison; n ≥ 60 cells per condition from 3 experiments; ns, non-significant).

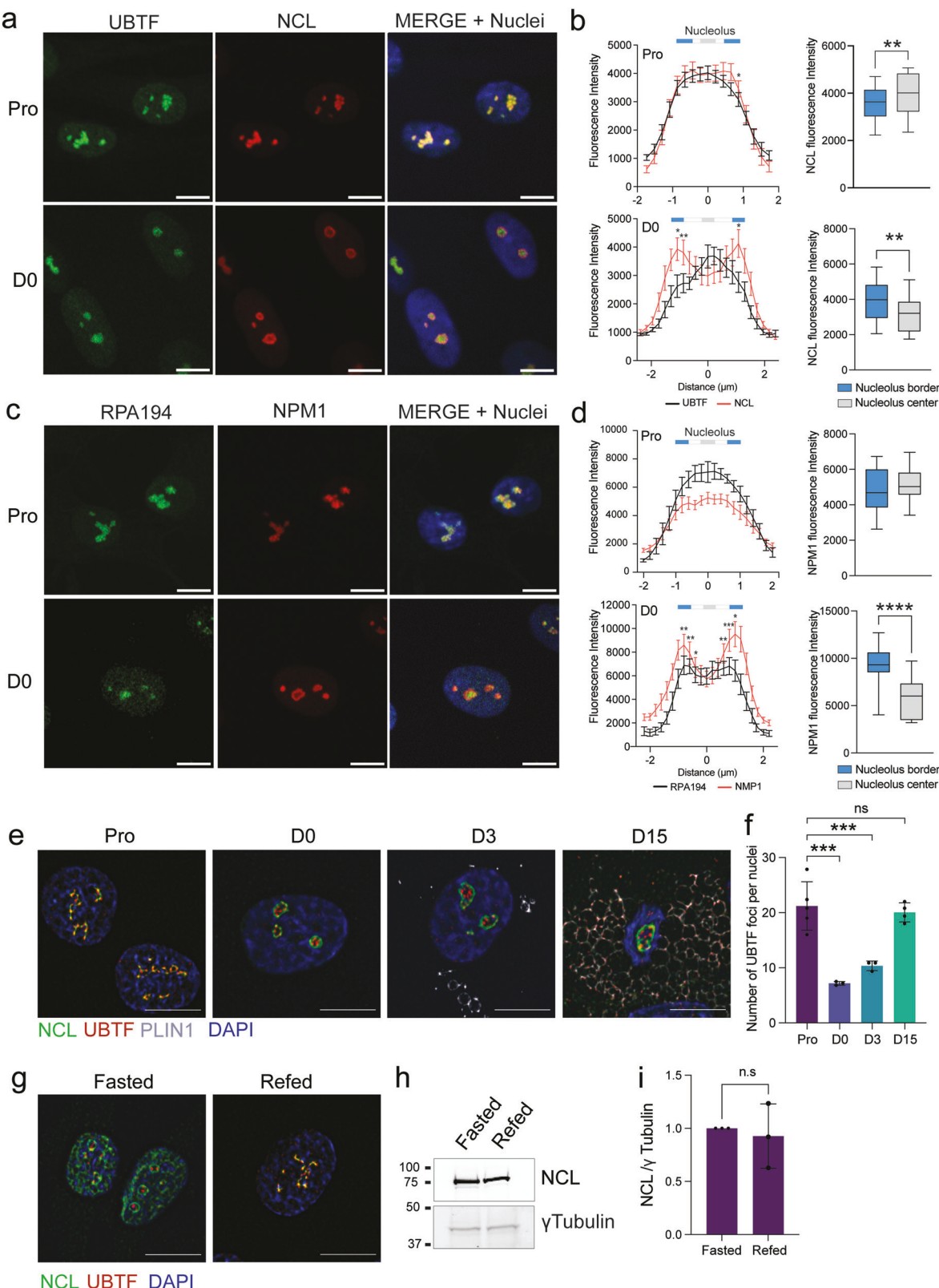

significantly lower than in proliferating ASCs. We next assessed protein synthesis capacity using a fasting and refeeding challenge (Fig. 4d–f; Supplementary Fig. 8b–d). While refeeding triggers the expected increase in translational activity in both proliferating and D0 cells, translational reactivation is significantly blunted at D15. This could relate at least in part to the increased protein levels of the translational inhibitor 4EBP1 (Fig. 4f, g;

Supplementary Fig. 8c). Hence, in the adipogenic differentiation system, increased nucleolar volume (see Fig. 2b) does not correlate with a significant increase in translational activity. Together with the morphological remodeling of nucleoli, these results suggest that nucleoli and the translational machinery undergo lineage-specific remodeling, leading to an altered nucleolar function in differentiated adipocytes.

**Fig. 3 | Cell cycle arrest triggers a rearrangement of nucleolar substructure.**
**a** Immunofluorescence analysis of Nucleolin (NCL) and UBTF in proliferating (Pro) and D0 conditions (scale bar: 10 μm). **b** Line profiles from (**a**) (left panel; *$p < 0.05$, **$p < 0.01$ vs nucleolus center, mixed effect analysis with Holm-Šídák's multiple comparisons; n ≥ 10 nucleoli per condition) and average NCL fluorescence intensity at nucleoli border vs center (right panel) (**$p < 0.01$ two-tailed paired *T* test; n ≥ 10 nucleoli per condition). **c** Immunofluorescence analysis of Nucleophosmin (NPM1) and RNA POL 1 (RPA194) in Pro and D0 conditions (scale bar: 10 μm). **d** Line profiles from **c** (left panel; *$p < 0.05$, **$p < 0.01$, ***$p < 0.001$ vs nucleolus center, mixed effect analysis with Holm-Šídák's multiple comparisons; n ≥ 10 nucleoli per condition) and average NCL fluorescence intensity at nucleoli border vs center (right

panel) (**$p < 0.01$ two-tailed paired *T* test; n ≥ 10 nucleoli per condition). **e** SRRF-Stream super-resolution microscopy images of Nucleolin and UBTF immunostainings in differentiating ASCs (scale bar: 10 μm). **f** Average number of UBTF foci per nuclei in differentiating ASCs (***$p < 0.001$ vs D0, one-way ANOVA with Tukey's multiple comparison; n ≥ 3 fields per condition from two independent experiments). **g** Representative SRRF-Stream super-resolution microscopy images of Nucleolin and UBTF immunostainings in fasted and refed undifferentiated ASCs. **h** Western blot analysis of NCL expression in fasted and refed conditions. γTubulin is shown as a loading control. **i** NCL signals normalized to γTubulin, quantified from western blots (non-significant (n.s), Wilcoxon matched-pairs signed rank test; *n* = 3). Data are presented as mean ± SD.

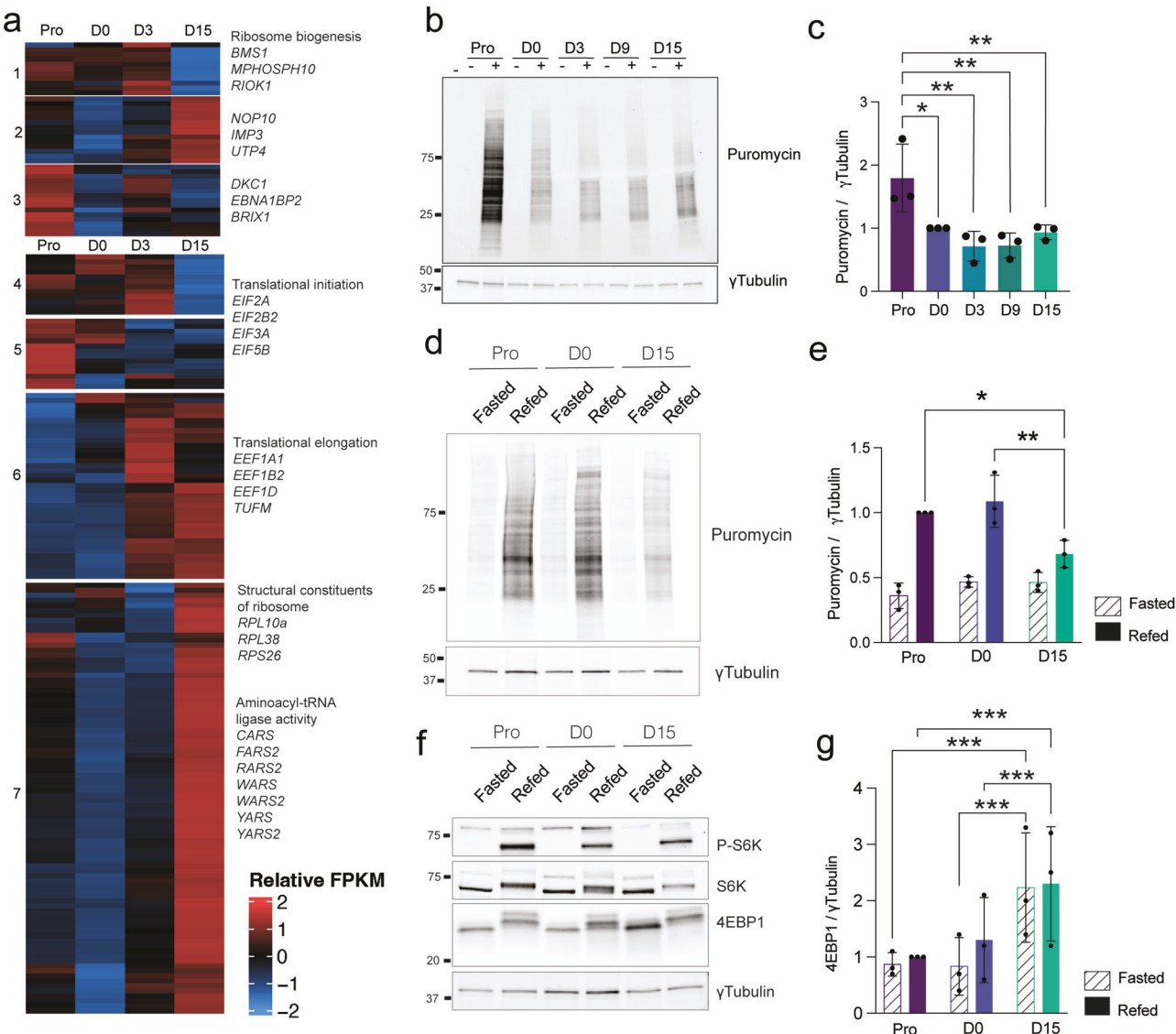

**Fig. 4 | Translation-related genes expression and translation efficiency during adipogenesis. a** Heatmap representation of relative gene expression (log2 FPKM) for genes pertaining to the ribosome biogenesis (GO:0042254) and translation (GO:0006412) gene ontology pathways (differentially expressed genes across time, *p* < 0.01, eBayes method, limma package). Selected genes of interest are highlighted for each cluster. **b** SUnSET analysis of protein synthesis during adipogenesis with (+) or without (−)1 μM Puromycin. γTubulin is shown as a loading control. **c** SUnSET signals normalized to γTubulin, quantified from (**b**) (*$p < 0.05$, **$p < 0.01$, two-way ANOVA with Tukey's multiple comparison test; *n* = 3).

**d** SUnSET analysis of protein synthesis rates in fasted and refed conditions during adipogenesis. γTubulin is shown as a loading control. **e** SUnSET signals normalized to γTubulin, quantified from (**d**) (*$p < 0.05$, **$p < 0.01$, two-way ANOVA with Sidak's multiple comparisons test; *n* = 3). **f** Western blot analysis of P-Thr389 S6K, total S6K and 4EBP1 in fasted and refed conditions during adipogenesis. γTubulin is shown as a loading control. **g** 4EBP1 signals normalized to γTubulin, quantified from western blots (***$p < 0.001$, two-way ANOVA with Sidák's multiple comparisons test; *n* = 3). Data are presented as mean ± SD.

## Cytoskeletal remodeling affects nuclear and nucleolar dynamics in adipose stem cells

Several lines of evidence suggest that nucleolar organization is mechanically regulated downstream of the actin cytoskeleton and nuclear remodeling[23,25,35]. However, our findings thus far suggest that cytoskeletal, nuclear, and nucleolar remodeling events occur with different kinetics during adipogenesis (see Figs. 1 and 2).

We, therefore, asked whether the nucleolus of differentiating ASCs is actively remodeled by changes in cytoskeletal tension, or passively by fusion due to decreased nuclear volume or chromatin compaction. We treated cells with increasing concentrations of Cytochalasin D (CytoD) to inhibit actin polymerization[37], with Jasplakinolide to stabilize actin cytoskeleton, or with Methylstat to promote heterochromatin accumulation through inhibition of histone H3K9me3 and H3K27me3 demethylases (KDM4C and 4E, and KDM6B respectively)[38] (Fig. 5a, b; Supplementary Fig. 9a, b). In ASCs, only the higher concentration of CytoD (5 µM) significantly perturbs actin cytoskeleton organization (Fig. 5a, c), while Jasplakinolide treatment increases actin signal intensity without affecting actin cytoskeleton organization (Fig. 5b, d). Both concentrations of CytoD, as well as Methylstat treatments, significantly reduce nuclear volumes, while Jasplakinolide treated cells tend to have a higher nuclear volume as previously reported (Fig. 5e)[13]. However, none of the compounds has a significant impact on cell area (Fig. 5f). Only CytoD treatments promote nucleolar fusion and formation of a single nucleolus, with a significant effect for the higher concentration of CytoD (Fig. 5a, g), confirming that nucleolar fusion events are controlled by cytoskeletal tension[35]. However, while nuclear and nucleolar volumes show a significant linear correlation in control, Jasplakinolide and Methylstat conditions, only actin depolymerization by a higher concentration CytoD leads to a significant increase in the No/Nu ratio (Fig. 5h,i; Supplementary Fig. 10). Thus, an organized F-actin network is required to maintain the relationship between nuclear and nucleolar sizes. We conclude that cytoskeleton disassembly is the main driver of nucleolar remodeling during adipogenesis.

## Discussion

We describe an adipose differentiation-driven remodeling of the nucleolus, that entails modifications of the structure, number, and volume of the most prominent nuclear organelle. While structural changes correlate with cell cycle arrest and reduced translational activity, the formation of a single nucleolus in differentiated adipocytes is an active mechanism regulated by actin cytoskeleton disassembly. Our results point to cytoskeletal dynamics as an important regulator of nucleolar remodeling towards the adipose cell fate.

Remodeling of the actin cytoskeleton and extracellular matrix is crucial for the commitment and differentiation of ASCs into adipocytes[5,39,40]. Indeed, drug-induced stabilization of the actin cytoskeleton is detrimental to adipogenesis while cytoskeletal disassembly enhances adipogenesis[13,41]. Recent studies from our group and others point to a functional link between cytoskeleton remodeling and translation regulation during adipogenesis[9,10]. Strikingly, disruption of these coordinated processes results in lipid storage defects in mature adipocytes, highlighting the importance of fine-tuned nucleolar function for the adipogenic differentiation program[9,10].

We find that growth arrest induced by contact inhibition or fasting triggers a structural change of nucleoli, characterized by a decrease in the number of UBTF-marked nucleolar foci and an enrichment of GC markers at the nucleolar periphery, reminiscent of the recently described nucleolar rim subcompartment[21]. This structural remodeling is not driven by actin cytoskeleton dynamics, and rather represents a feature of non-dividing, differentiating cells. Indeed, a similar structural rearrangement has recently been described during myogenic differentiation[42]. The distinct nucleolar structure of proliferating ASCs may therefore be related to the frequent re-assembly of nucleoli after cell division.

Our data indicate that, during adipogenesis, nucleolar morphology rather than number or volume correlates with translational activity. Altered nucleolar morphology during differentiation likely results from a decrease in nucleolar activity (i.e., reduced synthesis of ribosomal RNA by Pol I in agreement with decreased number of FCs and lower translation rate) since ribosomal RNA dictates nucleolar morphology and material properties of nucleoli[43–45]. Indeed, acute inhibition of Pol I by actinomycin D and other acute stresses result in pronounced reduction of nucleolar volume and translocation of GC proteins to the nucleoplasm[46]. However, nucleolar remodeling in differentiating ASCs does not result from acute cellular stress but instead, highlights the extent of variation of nucleolar morphology in physiological processes of cell division and differentiation.

The observed decrease in translational capacity following adipogenic induction could partly result from actin cytoskeleton disassembly[47]. Surprisingly, blunted translation rates in late differentiation are accompanied by a sharp increase in the transcription of ribosomal protein genes. Recent evidence points to ribosome heterogeneity as a key regulatory mechanism for stem cell differentiation[48]. The heterogeneity in ribosomal protein composition can notably provide ribosomes with a selectivity for translating transcripts involved in metabolism and development[49]. Such specialized ribosomes could allow a cell type-specific translational control, regulating the proteome during differentiation[50]. Transcriptional induction of various components of the translation machinery could also relate to extra-ribosomal roles of these proteins. Of note, eukaryotic translation initiation factor 2a (eIF2a) and 4e (eIF4e), both upregulated in late adipogenesis, have been shown to regulate lipid metabolism and insulin sensitivity[51–53], key functions of adipocytes[54,55].

Both nuclei and nucleoli undergo a drastic remodeling during adipogenesis: nuclear volume is reduced by half by D15 in our system, and the nucleus contains a single round nucleolus at this stage. In contrast to a previous report[23], we find that the number of nucleoli does not scale with nucleus size during adipogenesis. The decrease in nucleolus number precedes the change in nucleus volume in our system, suggesting an active remodeling mechanism rather than a passive fusion of nucleoli merely due to reduced nuclear volume. In line with this, the reduction of nuclear size triggered by Methylstat treatment does not result in the fusion of nucleoli. Methylstat promotes the accumulation of heterochromatin, which in turn, was reported as an unfavorable environment for fusion of nuclear condensates[56], such as nucleoli. In contrast, we report that disruption of actin filaments (F-actin) by CytoD promotes the formation of a single round nucleolus in dividing cells. These results add to emerging evidence that the nucleolus is a mechanosensitive organelle in human cells[23–25]. Interestingly, CytoD disrupts the linear correlation between nuclear and nucleolar sizes in proliferating ASCs, pointing to the importance of a preserved cytoskeleton for size scaling of the nucleolus.

Adipogenesis is a mechanically dynamic system where intracellular tensions vary as the cellular microenvironment is remodeled, the cytoskeleton is reshaped, and stiff lipid droplets accumulate[13,57]. In vivo, adipose tissue expansion is also under control of mechanical cues[58]. In fact, altered adipose tissue biomechanics are a hallmark of adipose tissue pathologies such as obesity or lipodystrophic syndromes. As obesity is associated with a deregulation of genes linked to tissue remodeling and protein translation[59], one may speculate that altered nucleolar functions might also be at play in obesity-associated adipose tissue dysfunction.

## Methods

### Cell culture and differentiation

Primary ASCs were isolated from subcutaneous fat obtained by liposuction from two unrelated female donors after informed consent was given (Donor 1: 50 years old, BMI 27.4; Donor 2: 45 years old, BMI 20,1; approval by the Regional Committee for Research Ethics for Southern Norway, REK 2013/2102 and 2018-660). ASCs were cultured in DMEM/F12 (17.5 mM glucose) with 10% fetal calf serum and 20 ng/ml basic fibroblast growth factor (proliferation [Pro] medium). Upon confluency, fibroblast growth factor was removed, and cells were cultured for 72 h in DMEM/F12 (17.5 mM glucose) with 10% fetal calf serum (basal medium) before induction of differentiation [D0]. For adipose differentiation, ASCs were induced with a

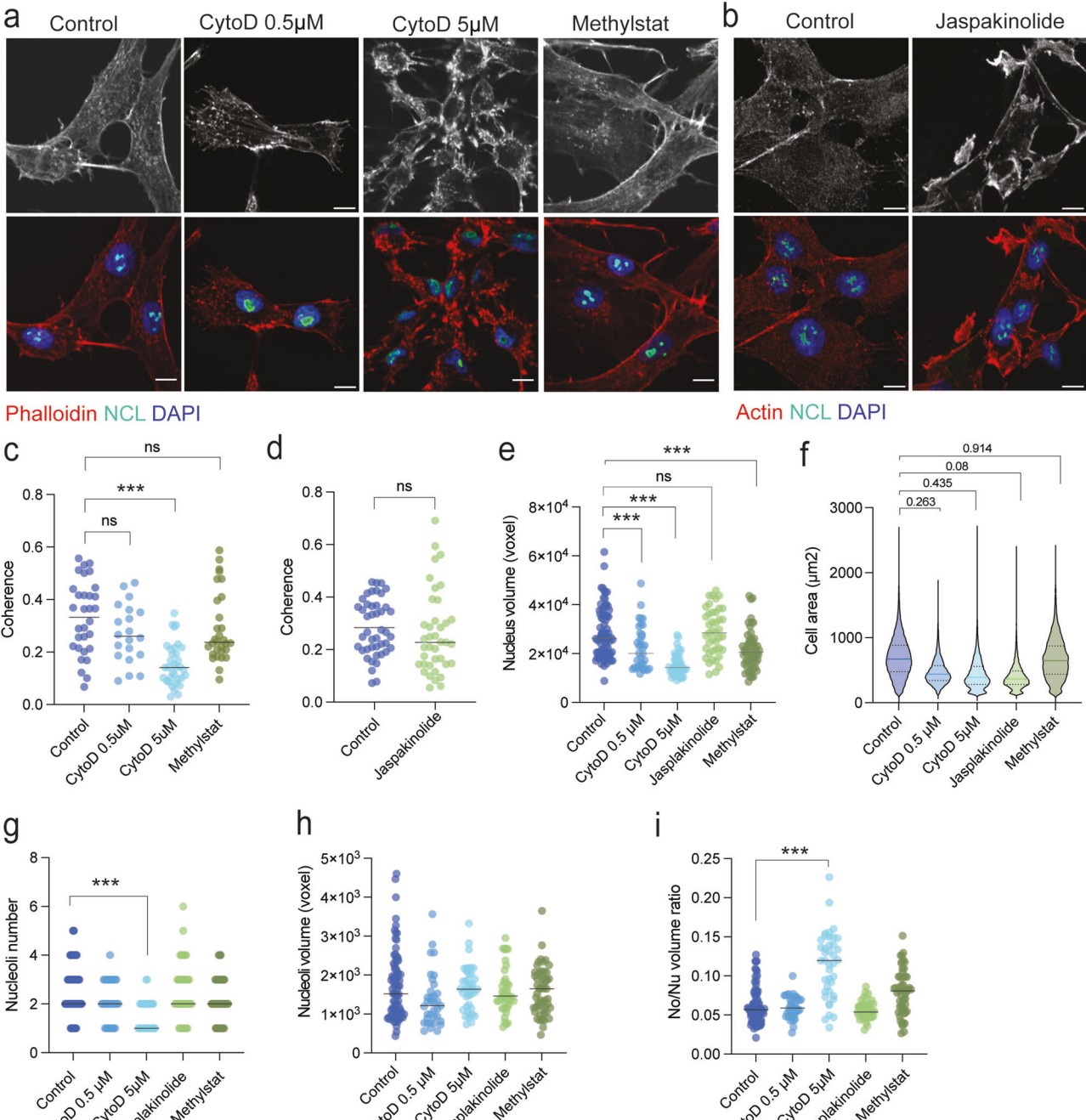

**Fig. 5 | Cytoskeleton structure defines nucleoli number. a** Phalloidin staining only (upper panel) or merged Nucleolin (NCL) immunostaining with DAPI (lower panel) in control condition, and after cytochalasin D (CytoD) 0.5 μM or 5 μM, and Methystat 1 μM treatments in proliferating ASCs (scale bar: 10 μm). **b** Actin immunostaining only (upper panel) or merged Nucleolin (NCL) immunostaining with DAPI (lower panel) in control condition, and after Japakinolide 2 μM treatment (scale bar: 10 μm). **c, d** Coherence measurements from (**a**) and (**b**), respectively (***$p < 0.001$ vs Control, Two-way ANOVA with Tukey's multiple comparison test; n ≥ 20 cells from 3 independent experiments). **e** Nucleus volume (***$p < 0.001$ vs Control, Two-way ANOVA with Tukey's multiple comparison test; n ≥ 35 cells from 3 independent experiments) and (**f**) cell surface (pairwise $t$-test with holmberg adjustment for multiple testing; ≥10,000 cells from 3 independent experiments) in control condition, and after CytoD 0.5 μM or 5 μM, Methystat 1 μM and Japakinolide 2 μM treatments. **g** Nucleoli number, (**h**) nucleoli volumes and (**i**) nucleolus-to-nucleus (No/Nu) volume ratio measured from Nucleolin and DAPI in control condition, and after CytoD 0.5 μM or 5 μM, Methystat 1 μM or Jasplakinolide 2 μM treatments (****$p < 0.0001$ vs Control, Two-way ANOVA with Tukey's multiple comparison test; n ≥ 35 cells from 3 independent experiments).

cocktail of 0.5 μM 1-methyl-3 isobutyl xanthine, 1 μM dexamethasone, 10 μg/ml insulin and 200 μM indomethacin in basal medium. For osteogenic differentiation, ASCs were induced with 0.1 μM dexamethasone, 10 mM β-glycerophosphate and 0.05 mM L-ascorbic acid-2 phosphate in basal medium. Differentiation media was renewed every 3 days, and samples were harvested 1 [D1], 3 [D3] and 15 days after induction [D15]. For mature adipocyte isolation, D15 samples were trypsinized, resuspended in HBSS,

0.5% BSA, and centrifuged 200 g for 5 min. Floating mature adipocytes were lysed in Trizol and stored at -80°C until RNA extraction. All differentiation experiments were done in at least three biological replicates between passages 4 and 9. For growth arrest experiments, proliferating ASCs were maintained in HBSS containing 1 g/L glucose for 24 h (Fasted) then switched back to Pro medium for 24 h (Refed). To trigger cytoskeletal remodeling, proliferating ASCs were treated with Cytochalasin D (Sigma, C8273)

5 μM for 30 min or 0,5 μM for 1 h, or with 2 μM Jasplakinolide (Sigma, 420127) for 30 min. To promote heterochromatin accumulation, cells were treated with 1 μM Methylstat (Sigma, SML0343) for 48 h.

## Oil red O staining

D15 cells were fixed in 4% paraformaldehyde for 20 min then stained for 30 min in Oil Red O (O0625; Sigma-Aldrich) diluted in isopropanol.

## BrdU labeling

Pro, Fasted, Refed and D0 ASCs were incubated in their respective medium for 8 h before addition of BrdU (10 μM final concentration) and further incubation for 18 h. Controls without BrdU were performed for each condition. Cells were then processed for BrdU and DNA labeling using an APC BrdU kit (BD pharmingen, BDB552598) following manufacturer's instructions. The flow cytometry analyses were performed on a NovoCyte (Acea Biosciences Inc.). All gatings are presented in Supplementary Fig. 11.

## Cell surface measurements

Cells were treated as indicated then washed three times in PBS before fixation in situ with 4% paraformaldehyde for 10 min. Cells were then trypsinized for 10 min and harvested in PBS containing 1% bovine serum albumin (BSA, Sigma A7906). Cells were incubated with 2 μg/ml Hoechst 34580 (Sigma 63493) for 15 min, washed 3 times in PBS and resuspended in 50 μL PBS 2% BSA. Samples were loaded on an ImageStreamX Mk II flow imager (Amnis) and imaged using a 40× objective and the following channels: (i) Ch01, Brightfield (430–480 nm) (ii) Ch07, Hoechst (405 nm/430–505 nm) (iii) Ch06, Darkfield SSC (740–800 nm). An initial gating was performed to exclude debris and cells out of focus (Brightfield gradient RMS < 45) before imaging 10000 cells per sample (Supplementary Fig. 12a, b). Downstream analysis was performed using the IDEAS 6.0 software using default masking options for brightfield (Ch01/M01) and Hoechst (Ch07/M07) (Supplementary Fig. 12c, d). Single cells were gated based on Hoechst intensity after gating out cells with out of focus nuclei (Supplementary Fig. 12e, f). The same gating was applied to all samples and image sorting based on Hoechst was then performed to ascertain the absence of doublets in the "single cells" population. A repeated measures ANOVA was conducted on the median area of all cells per replicate ($n = 3$), with a post-hoc pairwise $t$-test with holmberg adjustment for multiple testing.

## Immunofluorescence

Cells were grown on 12-mm diameter coverslips in 24-well plates. On the day of fixation, cells were washed 3 times with PBS, fixed with 4% ice-cold paraformaldehyde or methanol for 10 min, washed 3 times, and stored in PBS at 4 °C for up to one week. For staining, cells were permeabilized with PBS containing 0.01% Triton X-100, 0.01% Tween 20, and 2% BSA. Coverslips were then incubated with primary antibodies listed in Table S1 and secondary antibodies sequentially for 1 h at room temperature. DNA was stained with 2 μg/ml Hoechst 33342 (ThermoFisher, 62249) for 15 min and actin cytoskeleton with Phalloidin 594 (ab176757, Abcam) for 15 min. Coverslips were mounted in DAKO Fluorescence Mounting Medium (S3023, Agilent).

## Immunoblotting

Proteins were resolved by gradient 4-20% SDS–PAGE, transferred onto nitrocellulose membranes (BioRad) and blocked with 5% BSA. Membranes were incubated overnight using antibodies listed in Table S1. Proteins were visualized using IRDye-800-, IRD IRDye-680-, or HRP-coupled secondary antibodies. The surface sensing of translation (SUnSET) assay[36] was done by labeling newly synthesized proteins with 1 μM Puromycin for 1 h in steady-state conditions, or following 24 h starvation (HBSS 1 g/L glucose, 1% fatty acid-free BSA) and 24 h refeeding (basal medium). Translation rates were analyzed by western blotting using anti-Puromycin antibodies. Bands were quantified by densitometry (Image Lab; BioRad) using γTubulin for normalization. Uncropped membranes are presented in Supplementary Figs. 12–19.

## RNA-sequencing

RNA sequencing (RNA-seq) was done in biological triplicates for 2 independent donors. Total RNA was isolated from samples harvested at four differentiation time-points (Pro, D0, D3, D15), as well as from mature adipocytes (ADI) isolated from D15 samples using the RNeasy kit (QIAGEN). PolyA-selected RNA was sequenced from paired-end libraries (TruSeq Stranded mRNA kit; Illumina) using NextSeq or Novaseq platforms (Illumina). Reads were aligned to the hg38 genome (ensembl95 annotation) with hisat2[60] and counted using featureCounts (-fraction -M)[61]. To produce a list of genes that change expression over time, normalized read counts were compared between consecutive time-points and tested in combination to provide FDR-adjusted $p$-values (significance was attained at $p < 0.01$ using Limma's robust eBayes method)[62]. Explicitly, Pro-D0, D0-D3 and D3-D15, were compared for each donor.

Gene Set Variation Analysis[63] was used to calculate an enrichment score for average gene expression in the gene lists for Hallmark Adipogenesis (M5905[64]) and Gene Ontology (GO) terms Mitotic Cell Cycle Arrest (GO:0071850) and Mitotic cell cycle (GO:0000278) (MSigDB v7.4). Heatmaps were generated for genes in GO terms Translation (GO:0006412), Ribosome biogenesis (GO:0042254) and Cytosolic ribosome (GO:0022626), and scaled log2 RPKM. Each gene was hierarchically clustered using Euclidean distance and the complete linkage method. Based on visual inspection of heatmaps, dendrograms (not shown) were cut to produce 3 or 4 clusters.

## Microscopy and image analysis

Phase contrast images were taken on an Olympus CKX53 microscope using 10× and 20× objectives and an Olympus UC90 camera. All confocal microscopy images were taken with a Dragonfly 505 spinning disk microscope equipped with a Borealis perfect illumination delivery system and a CFI apochromat (Oxford Instruments, iXon Life 888 EMCCD camera, 60 × 1.4 NA objective, 4-μm stack with 0.3 μm steps). Laser/filters were as follows: (i) far red 637 nm/700–775 nm (ii) red 561 nm/620 nm (iii) green 488/525 (iv) blue 405 nm/450 nm. The volume of nuclei and nucleoli, and the number of nucleoli per cell were measured by Nemo[65] using the Nucleolin and Hoechst channels to segment nucleoli and nuclei, respectively. Lipid droplets were measured using Morphological Segmentation tool from MorphoLibJ[66]. Nuclear shape was evaluated using NucleusJ[67]. Actin cytoskeleton coherence was measured from phalloidin staining with the OrientationJ plugin (http://bigwww.epfl.ch/demo/orientation/) in ImageJ (National Institutes of Health). MiNa plugin[68] was used to measure branch length and total volume occupied by cytoskeleton from full z stacks as previously described[9]. UBTF foci quantification was done with Imaris (Oxford Instruments) software, the number of UBTF foci was calculated per field and divided by the number of cells in the field to achieve an average per cell. Intensity profiles were generated in ImageJ (see Table S2). SRRF stream images were taken using the same objective and slides as above.

## Statistics and reproducibility

Statistical analysis of images and western blots quantifications were performed with GraphPad Prism 8.4.2 (GraphPad Software). Statistics were performed on $n = 3$ biological replicates, defined as independent differentiation experiments. Data were plotted as mean ± standard deviation. For image analyses, technical replicates (individual cells) from all 3 biological replicates were plotted. Sample size for each experiment is indicated in the figure legends. Gaussian distribution of data was tested using the Shapiro-Wilks test. We used one-way ANOVA or mixed-effect analysis with Tukey's or Holm-Šídák's multiple comparisons tests (95% confidence interval) to evaluate the significance of the difference in mean values between time points. When applicable, we used two-way ANOVA with Šídák's multiple comparisons test.

## Reporting summary

Further information on research design is available in the Nature Portfolio Reporting Summary linked to this article.

## Data availability

Previously published RNA-seq data are available from Gene Expression Omnibus (GEO) accession GSE176020, whilst new data for D15 adipocytes are available at accession GSE227819. Datasets from Rauch et al. are available from GEO accession GSE113253. The numerical source data behind the graphs can be found in the Supplementary data file. All other data are available from the corresponding author on reasonable request.

## Code availability

Code for data processing and analysis is available at https://github.com/sarahhp/nucleolus_paper.

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

## Acknowledgements
This work was funded by South-East Health Norway (grant 40040) and the Research Council of Norway (grants 249734 and 313508). We acknowledge Michael Daws for his help with flow imaging experiments, and the Norwegian Sequencing Center (Oslo University Hospital) for professional services.

## Author contributions
E.P. and N.B. designed the study, performed experiments and analyzed data. S.H.P. did bioinformatics analysis. A.B. performed experiments. T.G. contributed to experimental design and microscopy analysis. E.P., S.H.P. and N.B. made figures. E.P. and N.B. wrote the manuscript. S.H.P., A.B., T.G. and P.C. revised the manuscript. NB is a supervisor and guarantor of this work.

## Competing interests
The authors declare no competing interests.
