## [Peer Review File · Communications Biology]

Reviewers' comments:

Reviewer #1 (Remarks to the Author):

The nucleolus is a biomolecular condensate formed by liquid-liquid phase separation where the initial steps of ribosome biogenesis take place. Although the nucleolus has been the subject of intense research for many years, it has remained unclear how mechanobiology cues, such as compressive forces, influence its morphology and function. In this work the authors have used a model of adipocyte differentiation to start to address some aspects of it.

Adipocyte maturation is characterized by formation and accumulation of lipid droplets in the cytoplasm and a concomitant reorganization of the actin cytoskeleton and cell nucleus. The question was to know if the nucleolus is also remodeled, the answer is clearly: yes. The internal organization of the nucleolus is changed (relative distribution of subnucleolar domain antigens), the number of nucleoli decreases (nucleolar fusion), the shape of the nucleolus also changes (more spherical), and its respective volume increases (nucleolar-to-nuclear ratio). The effects on the nucleolus are not just a consequence of the impact on the nucleus because the trends are different, and there is a lineage-specificity component as cells engaged in osteogenic differentiation don't show the same effects. Treating cells with an actin depolymerizing drug recapitulates many of the observed effects on nucleolar structure indicating the involvement of an active process in this mechanobiology response/sensing. Note that osmolarity shock also leads to nucleolar fusion but in this case the nucleolar-to-nuclear ratio is not changed indicating turgescence alone, such as the one expected from cytoplasmic lipid droplet growth, is not enough. Some functional consequences have also been addressed (this is more preliminary), including: effects on translation (by assessing global a.a. incorporation), and somehow on ribosome biogenesis (but this is rather indirect, relying on expression assessment of trans-acting factors and ribosomal protein in RNAseq data; overall there is less information to learn from this part of the work).

In conclusion, this is a nice, original work. The experiments have been well performed and described. The narrative is good, it's a pleasant read. Of course, it would be better to sustain further the functional consequences on nucleolar metabolism, and, in particular, to provide more direct proofs of an effect (or not) on ribosome biogenesis but this could be pursued in the future.

Note, in the Discussion, a comparison with the 'nucleolar rim' (a recently described peripheral layer of the nucleolus, and very specific structure) is made. If the authors want to do this, they should colocalize components with known nucleolar rim marker proteins.

Also note, a new layer between the DFC and GC has recently been identified by super resolution microscopy, and coined 'the periphery of dense fibrillar component (PDFC)'. It would be ideal if the authors could include this information in their Introduction to be up-to-date. In particular since NCL, used in this work, turned out to be a PDFC antigen.

It would be useful to discuss briefly why the authors think there is a diphasic variation in total nucleolar

volume during differentiation.

Reviewer #2 (Remarks to the Author):

In their manuscript, Potolitsyna et al. investigate the morphological remodeling of the nucleus and nucleolus during in vitro adipogenic differentiation of primary human adipose stem cells (ASCs). They report that cell cycle arrest of primary human ASCs results in nucleolar remodeling in a manner that correlates with a decrease in protein synthesis. Moreover, the authors reveal that treating these cells with cytochalasin D or sorbitol mimics the nucleolar remodeling observed during adipogenesis. Based on these results, Potolitsyna conclude that nucleolar remodeling during adipogenic differentiation is an active, mechano-regulated process that is likely governed by the actin cytoskeleton. While the findings reported in this manuscript are intriguing, it is perhaps premature for the authors to conclude a critical role for actin cytoskeletal rearrangements in defining nuclear and nucleolar architecture in differentiating human adipose stem cells. There are several major and minor issues that significantly reduce my enthusiasm for this manuscript in its current form. Thus, I cannot recommend that this manuscript be published in its current form until the authors address my concerns, which are briefly outlined below.

Major issues:

1) Figure 1:

- a. Panels A and E: It would be helpful if the authors were to provide more images of the time points between D3 and D15, to better illustrate the nucleus undergoing massive morphological changes during this time. It would also be appreciated if the authors were to provide non-merged images in addition to the merged images that they presented in their original manuscript. This will help their readers better interpret the authors' results.
- b. Panel B: Quantifying the "coherence" of actin in proliferating, D0, D3, and D15 ASCs is one way of measuring the overall organization of the cytoskeleton. However, it would be helpful if the authors were to provide more information about the organization of the actin cytoskeleton during ASC differentiation. For example, the authors could measure the average fluorescent intensity of phalloidin-stained actin in these cells and the number of actin cables above their nuclei.
- c. Panel C: The use of γ -tubulin to normalize the levels of lamin A in differentiating ASCs feels strange to me given that differentiated cells, like adipocytes, are known to form non-centrosomal microtubule organizing centers on their nuclear envelopes. Perhaps the authors could repeat this experiment with a less potentially problematic control protein?
- d. Panel D: It would be helpful if the authors were to report the volume of their cells in addition to their nuclear volume.

2) Figure 2:

- a. Panels A, C, E, and G: It would be appreciated if the authors were to provide non-merged images in

addition to the merged images that the presented in their original manuscript. This will help their readers better interpret the authors' results.

b. Panels B and D: To better support their conclusions, the authors might consider providing average line scans from a statistically-appropriate number of cells rather than line scans from one nucleolus in one nucleus.

c. Panels E and F: It seems like there are numerous UBTF foci in the cytoplasm of the D15 cells. Were these foci considered in the quantification reported in Panel F? What are these PLIN1-associated UBTF foci?

d. Panel G: Does the increased nucleoplasmic NCL levels in the fasted cell presented here correlate with an increased amount of NCL expression globally?

3) Figure 3:

a. It seems like a missed opportunity for the authors not to have tested the effect of cytochalasin D, sorbitol, or methylstat on the expression of translation-related genes and translation efficiency during adipogenesis.

4) Figure 4:

a. Panel A:

i. Some quantification of the disruption of the actin cytoskeleton would be appreciated here, as the extent of actin depolymerization is not entirely clear. Perhaps the authors could return to the coherence metric utilized in Figure 1?

ii. The use of 5 μM cytochalasin D is rather excessive, as demonstrated by the massive cytomorphological effects exerted by the drug on the cells shown here. Can the authors try to repeat the experiment with less cytochalasin D (e.g., 0.5 μM) for 1 hour to see if they might be able to find a condition where the actin cytoskeleton is depolymerized in the absence of major changes in cellular volume? In addition, it would be useful for the authors to show the effects of stabilizing the actin cytoskeleton or activating actomyosin contractility by treating cells with jasplakinolide or calyculin A, respectively. It would be good to know if these treatments have the opposite (or not) effect on nucleoli during adipogenesis.

iii. Sorbitol does much more than "remodel the actin cytoskeleton", as stated by the authors. For example, the hyperosmotic stress exerted on cells by their exposure to sorbitol will result in cellular shrinkage and increased cytoplasmic macromolecular crowding, which will impact numerous biochemical reactions. Therefore, the authors cannot conclude that the effects that they are observing in sorbitol-treated cells are due exclusively to remodeling of the actin cytoskeleton.

iv. A control for methylstat is needed to be able to conclude that it is working properly in these cells.

b. Panels B-E: The authors should report the volume of their cells in addition to their nuclear and nucleolar volumes.

5) Supplementary Figure 4:

a. Panel A: It is unclear which of these pictures represents the D0 ASC, the D15 ASC in basal medium, or the D15 ASC in osteogenic differentiation medium. The authors should label the images accordingly.

Minor issues:

1) Page 4, Line 19; Page 6, Lines 6 and 11: The authors should insert “actin” before “cytoskeleton” for specificity.

2) Page 5, Line 7: The authors need to define what the “SUnSET assay” is, how it is used to assess the levels of protein synthesis, and to provide a reference for this technique when they first introduce it.

3) Page 5, Lines 26-28: The authors need to provide a reference for “methylstat” here.

4) Page 9, “Confocal microscopy and image analysis” section of the Materials and Methods: The authors should provide information about the following: the exact EMCCD camera used, the light source used, and the excitation/emission filters used.

5) Figures 3B, 3D, 3F, S1C, S2A, S7A, S7B, S7C, S9, S10, S11, S12, S13: The authors need to provide the positions of their molecular weight standards in all the Western blots presented in this manuscript.

6) Figure 4A: “Methstat” should be changed to “Methylstat”.

Reviewer #3 (Remarks to the Author):

In this manuscript, Evdokiia and colleagues have investigated the role of the actin cytoskeleton in shaping the nuclear and nucleolar architecture during adipogenesis. They used primary human adipose stem cells and analyzed the nuclear and nucleolar remodeling process of adipogenesis. Although nucleolar structure appears to be correlated with translational activity during adipogenesis, additional experiments are required to determine whether cytoskeletal remodeling is essential for nucleolar architecture-mediated adipogenesis. Followings are specific comments to improve the study:

1. Even though the authors identified the nucleolar dynamics during adipogenesis and demonstrated a correlation between nucleolar changes and several time points of adipogenesis, they should carry out experiments that manipulate nucleolar dynamics to validate their effects on adipogenesis. Otherwise, it is unclear whether the changes in the nucleolus would be a cause or consequence of adipogenesis. Moreover, the biological significance of nucleolar number, size, or sub-compartmental organization during adipogenesis should be provided.

2. Considering the previous report that actin-associated genes are altered upon changes in lipid droplet locularity (Kim et al., *Molecular and Cellular Biology* 39.20, 2019: e00210-19), the authors should investigate whether the enlarged and unilocular lipid droplet in adipocytes would affect cytoskeletal remodeling that results in nuclear and nucleolar changes.

3. The rationale to select day 3 for adipogenesis time point is unclear. Is it related to the periods of changes in the expression of adipogenic key regulators (PPARgamma, C/EBPalpha, etc.)? Also, is there any link between key adipogenic transcription factors and the number or shape of nucleolus? The authors should properly address these issues.

4. The authors should provide appropriate data representing adipogenesis. For instance, they should examine lipid droplet formation via Perilipin1 and BODIPY staining (Figs. 1, 2, 4, Supplementary Figs. 1, 4) and measure the expression levels of adipogenic marker genes. These data are crucial to proving that the observed nucleolus changes would be associated with adipogenesis.
5. In Fig. 2, the authors should examine markers whether the cell cycle has been clearly arrested. For example, they should investigate the expression levels of Cyclin, Cdk, etc., under each experimental condition, referring to previous results analyzing cell cycle arrest in adipocyte differentiation (Reichert et al., *Oncogene* 18.2, 1999: 459-466.).
6. The images of differentiating adipocytes mainly focus on the nucleus, making it difficult to recognize the cell's overall shape (Figs. 1a, 1e, Fig. 2, Supplementary Figs. 1a, 4a). Providing images taken at a lower magnification would be helpful.
7. Proper explanation is required for different changes in nucleolar volume observed in Fig. 1f and Supplementary Fig. 3b. It seems that they would be the result from the same experimental design.
8. In Fig. 4, the drug treatment conditions are not described. Cellular morphology of the control group in Fig. 4 looks like proliferating adipose stem cells. To examine the effect of the cytoskeleton on nucleolar dynamics observed during adipogenesis, it is recommended that drugs be administered at day 0, when adipose stem cells are reached confluence and are growth-arrested.

Reviewers' comments:

Reviewer #1 (Remarks to the Author):

The nucleolus is a biomolecular condensate formed by liquid-liquid phase separation where the initial steps of ribosome biogenesis take place. Although the nucleolus has been the subject of intense research for many years, it has remained unclear how mechanobiology cues, such as compressive forces, influence its morphology and function. In this work the authors have used a model of adipocyte differentiation to start to address some aspects of it.

Adipocyte maturation is characterized by formation and accumulation of lipid droplets in the cytoplasm and a concomitant reorganization of the actin cytoskeleton and cell nucleus. The question was to know if the nucleolus is also remodeled, the answer is clearly: yes. The internal organization of the nucleolus is changed (relative distribution of subnucleolar domain antigens), the number of nucleoli decreases (nucleolar fusion), the shape of the nucleolus also changes (more spherical), and its respective volume increases (nucleolar-to-nuclear ratio). The effects on the nucleolus are not just a consequence of the impact on the nucleus because the trends are different, and there is a lineage-specificity component as cells engaged in osteogenic differentiation don't show the same effects. Treating cells with an actin depolymerizing drug recapitulates many of the observed effects on nucleolar structure indicating the involvement of an active process in this mechanobiology response/sensing. Note that osmolarity shock also leads to nucleolar fusion but in this case the nucleolar-to-nuclear ratio is not changed indicating turgescence alone, such as the one expected from cytoplasmic lipid droplet growth, is not enough.

Some functional consequences have also been addressed (this is more preliminary), including: effects on translation (by assessing global a.a. incorporation), and somehow on ribosome biogenesis (but this is rather indirect, relying on expression assessment of trans-acting factors and ribosomal protein in RNAseq data; overall there is less information to learn from this part of the work).

In conclusion, this is a nice, original work. The experiments have been well performed and described. The narrative is good, it's a pleasant read. Of course, it would be better to sustain further the functional consequences on nucleolar metabolism, and, in particular, to provide more direct proofs of an effect (or not) on ribosome biogenesis but this could be pursued in the future.

We thank the reviewer for the positive feedback.

Note, in the Discussion, a comparison with the 'nucleolar rim' (a recently described peripheral layer of the nucleolus, and very specific structure) is made. If the authors want to do this, they should colocalize components with known nucleolar rim marker proteins.

In the original paper, the nucleolar rim is defined by "a characteristic rim-like pattern" similar to what we observe in our system (Stenström et al., Mol Syst Biol 2020). Nucleolin and NMP1 are both described as rim markers in the same study.

Also note, a new layer between the DFC and GC has recently been identified by super resolution microscopy, and coined 'the periphery of dense fibrillar component (PDFC)'. It would be ideal if the authors could include this information in their Introduction to be up-to-date. In particular since NCL, used in this work, turned out to be a PDFC antigen.

We thank the reviewer for the suggestion and have updated the introduction accordingly (l.51-53).

It would be useful to discuss briefly why the authors think there is a diphasic variation in total nucleolar volume during differentiation.

We observe an initial drop in total nucleoli volume when comparing proliferating ASCs (Pro) to confluent, growth arrested D0 cells. Nucleoli volume then increases as differentiation proceeds (D3,D15). This has now been clarified in the results section (l.89-92).

Reviewer #2 (Remarks to the Author):

In their manuscript, Potolitsyna et al. investigate the morphological remodeling of the nucleus and nucleolus during in vitro adipogenic differentiation of primary human adipose stem cells (ASCs). They report that cell cycle arrest of primary human ASCs results in nucleolar remodeling in a manner that correlates with a decrease in protein synthesis. Moreover, the authors reveal that treating these cells with cytochalasin D or sorbitol mimics the nucleolar remodeling observed during adipogenesis. Based on these results, Potolitsyna conclude that nucleolar remodeling during adipogenic differentiation is an active, mechano-regulated process that is likely governed by the actin cytoskeleton. While the findings reported in this manuscript are intriguing, it is perhaps premature for the authors to conclude a critical role for actin cytoskeletal rearrangements in defining nuclear and nucleolar architecture in differentiating human adipose stem cells. There are several major and minor issues that significantly reduce my enthusiasm for this manuscript in its current form. Thus, I cannot recommend that this manuscript be published in its current form until the authors address my concerns, which are briefly outlined below.

Major issues:

1) Figure 1:

a. Panels A and E: It would be helpful if the authors were to provide more images of the time points between D3 and D15, to better illustrate the nucleus undergoing massive morphological changes during this time. We have taken advantage of the diversity in lipid droplet content and size at D15 to address the impact of increased lipid storage on nuclear morphology, which is the point we are making. We now show significant correlations between nuclear volume, deformation and lipid droplet size (new Fig1. g,h,i). This new data clearly demonstrates that the mechanical load induced by the lipid droplet growth is the main driver for the massive morphological changes undergone by the nucleus in mature adipocytes.

It would also be appreciated if the authors were to provide non-merged images in addition to the merged images that the presented in their original manuscript. This will help their readers better interpret the authors' results.

We thank the reviewer for the suggestion and now provide images for each channel.

b. Panel B: Quantifying the “coherence” of actin in proliferating, D0, D3, and D15 ASCs is one way of measuring the overall organization of the cytoskeleton. However, it would be helpful if the authors were to provide more information about the organization of the actin cytoskeleton during ASC differentiation. For example, the authors could measure the average fluorescent intensity of phalloidin-stained actin in these cells and the number of actin cables above their nuclei.

Coherence was initially chosen because this metric does not depend on intensity values, allowing for comparison between D0 cells, for which phalloidin signal is very strong, and other time points. We now also provide a quantification of the total volume occupied by the actin cytoskeleton, as well as the average branch length of the network (Fig. 1 new panels b and d). Both measures reflect the progressive disruption of the actin cytoskeleton during adipogenesis.

c. Panel C: The use of α -tubulin to normalize the levels of lamin A in differentiating ASCs feels strange to me given that differentiated cells, like adipocytes, are known to form non-centrosomal microtubule organizing

centers on their nuclear envelopes. Perhaps the authors could repeat this experiment with a less potentially problematic control protein?

We respectfully disagree with the reviewer comment. Regardless of microtubule organization, the protein level of γ -tubulin is stable during differentiation, making it a valid and widely used loading control in adipogenic differentiation studies (Potelitsyna et al. *Sci. Rep.* 2022, Meissburger et al. *EMBO Mol Med* 2011, Gao et al. *Nature Comm.* 2013), including in studies focusing on lamin proteins (Oldenburg et al. *JCB* 2017, Madsen Østerbye et al. *Genome Biol.* 2022). Accordingly, and as shown here, reprobing of the membrane presented in Supplementary Fig. 2c with an antibody against GAPDH confirms that both loading controls display similar variations across experimental conditions.

d. Panel D: It would be helpful if the authors were to report the volume of their cells in addition to their nuclear volume.

Adipogenesis entails a dramatic increase in cell size to accommodate large lipid droplets in the cytoplasm. We have attempted to measure differentiating cells in two ways: (i) we first used a flow imager to assess cell surface. However, D15 cells are too large to be imaged even at the lowest magnification. (ii) we next attempted to measure cell volumes on confocal microscopy images. Because there is, to our knowledge, no cell surface marker that would be equally expressed in ASCs and mature adipocytes, we used a cytoplasmic membrane dye (Sigma, SCT109) to define individual cells. Due to varying levels of intracellular staining and/or punctuate patterns, we could not use these images for cell volume quantification (see panel below). Thus, accurate measurement of cell volume during *in vitro* adipogenesis is extremely challenging because of changes in cell size, shape and identity; we could not find such measurements in the published literature.

Membrane dye staining during adipogenic differentiation (scale bar: 20 μ m)

2) Figure 2:

a. Panels A, C, E, and G: It would be appreciated if the authors were to provide non-merged images in addition to the merged images that they presented in their original manuscript. This will help their readers better interpret the authors' results.

Non-merged images are now provided in the main figure for panels A and C, and as supplementary data for panels E and G (see new Fig. 3 and Supplementary Fig. 6).

b. Panels B and D: To better support their conclusions, the authors might consider providing average line scans from a statistically-appropriate number of cells rather than line scans from one nucleolus in one nucleus.

We thank the reviewer for the suggestion and now provide average line profiles for $n \geq 10$ nucleoli per condition (now Fig. 3b,d; left panels). Further, we have compared the average pixel intensities for Nucleolin and NPM1 at the nucleolus border vs nucleolus center (now Fig. 3b,d; right panels). These new data confirm that the GC proteins Nucleolin and NPM1 are significantly redistributed towards the periphery of the nucleolus in D0 cells.

c. Panels E and F: It seems like there are numerous UBTF foci in the cytoplasm of the D15 cells. Were these foci considered in the quantification reported in Panel F? What are these PLIN1-associated UBTF foci?

The cytoplasmic signal is due to the amplification of background fluorescence around lipid droplets by the SRFFstream algorithm. The number of UBTF foci has been quantified within nuclei, as now clarified in the figure legend (l.454) (now Fig. 3f) and in the method section (l.363-364).

d. Panel G: Does the increased nucleoplasmic NCL levels in the fasted cell presented here correlate with an increased amount of NCL expression globally?

New western blot data and quantifications indicate that redistribution of Nucleolin in fasted cells does not correlate with an increase in protein expression compared to the refed state (New Figure 3h,i).

3) Figure 3:

a. It seems like a missed opportunity for the authors not to have tested the effect of cytochalasin D, sorbitol, or methylstat on the expression of translation-related genes and translation efficiency during adipogenesis.

In our study, acute cytochalasin D treatment was used to address the role of cytoskeletal tension on nucleolar morphology. The effect of 24h Cytochalasin D treatment on gene expression in proliferating human ASC has been described elsewhere, with no effect on translation-related genes reported (Samsonraj et al. Stem cells dev 2018). Indeed, using their data, we find no overlap between Cytochalasin D-induced or -repressed genes and genes pertaining to the translation GO term (GO:0006412) (see Venn diagram). Since adipogenesis is accompanied by a decrease in F-actin (Chen et al. Stem cells research 2018), a stronger effect of Cytochalasin D treatment on gene expression later in differentiation seems unlikely.

The cytoskeleton associates with ribosomes, initiation factors and elongation factors, thereby regulating protein translation locally (Kim and Coulombe 2010). It is therefore likely that F-actin disruption in proliferating ASCs would impede translation, as previously described in other systems (Stapulionis et al. JBC 1997, Gross et al. Mol Cell Biol 2007, Silva et al. J. Cell Sci 2016). Thus, cytoskeleton disruption during adipogenesis could indeed contribute to the observed decrease in translational capacity. This is now discussed in the manuscript.

4) Figure 4:

a. Panel A:

i. Some quantification of the disruption of the actin cytoskeleton would be appreciated here, as the extent of actin depolymerization is not entirely clear. Perhaps the authors could return to the coherence metric utilized in Figure 1?

Coherence measurements based on Phalloidin or actin staining are now provided (see new Fig. 5a-d).

ii. The use of 5 mM cytochalasin D is rather excessive, as demonstrated by the massive cytomorphological effects exerted by the drug on the cells shown here. Can the authors try to repeat the experiment with less

cytochalasin D (e.g., 0.5 mM) for 1 hour to see if they might be able to find a condition where the actin cytoskeleton is depolymerized in the absence of major changes in cellular volume?

In addition, it would be useful for the authors to show the effects of stabilizing the actin cytoskeleton or activating actomyosin contractility by treating cells with jasplakinolide or calyculin A, respectively. It would be good to know if these treatments have the opposite (or not) effect on nucleoli during adipogenesis.

We thank the reviewer for the constructive comment. First, we would like to point out that our experiments were conducted using 5 μ M Cytochalasin D (*i.e.* 100X less than the suggested lower dosage). Based on reviewer's suggestions, we now present additional data including a treatment with the actin stabilizing agent Jasplakinolide, and a lower dose of Cytochalasin D (0.5 μ M for 1h). These data confirm that cytoskeleton disassembly is an important driver of nucleolar remodeling during adipogenesis.

iii. Sorbitol does much more than "remodel the actin cytoskeleton", as stated by the authors. For example, the hyperosmotic stress exerted on cells by their exposure to sorbitol will result in cellular shrinkage and increased cytoplasmic macromolecular crowding, which will impact numerous biochemical reactions. Therefore, the authors cannot conclude that the effects that they are observing in sorbitol-treated cells are due exclusively to remodeling of the actin cytoskeleton.

We agree. We present new data with both specific disruption and stabilization of the actin cytoskeleton that convincingly argue for a role of the cytoskeleton remodeling in regulating nucleolar morphology, and have chosen to remove the sorbitol treatment from our manuscript.

iv. A control for methylstat is needed to be able to conclude that it is working properly in these cells.

We have confirmed by western blotting of H3K27me3 and total H3 that Methylstat treatment promotes heterochromatin accumulation in our model (see new Supplementary Fig. 9a,b)

b. Panels B-E: The authors should report the volume of their cells in addition to their nuclear and nucleolar volumes.

Cell size has been measured in proliferating cells treated with cytochalasin D (0.5 μ M and 5 μ M) Jasplakinolide 2 μ M or Methylstat 1 μ M using an Amnis X100 flow imager ($n \geq 10\ 000$ cells per condition). We find that cell size is not significantly affected by the treatments used (see new Fig.5f).

5) Supplementary Figure 4:

a. Panel A: It is unclear which of these pictures represents the D0 ASC, the D15 ASC in basal medium, or the D15 ASC in osteogenic differentiation medium. The authors should label the images accordingly.

Done (now Supplementary Fig. 5).

Minor issues:

1) Page 4, Line 19; Page 6, Lines 6 and 11: The authors should insert "actin" before "cytoskeleton" for specificity.

Done.

2) Page 5, Line 7: The authors need to define what the "SUnSET assay" is, how it is used to assess the levels of protein synthesis, and to provide a reference for this technique when they first introduce it.

Done (l.156-157).

3) Page 5, Lines 26-28: The authors need to provide a reference for "methylstat" here.

Done.

4) Page 9, "Confocal microscopy and image analysis" section of the Materials and Methods: The authors should provide information about the following: the exact EMCCD camera used, the light source used, and the excitation/emission filters used.

Done (l.351-355).

5) Figures 3B, 3D, 3F, S1C, S2A, S7A, S7B, S7C, S9, S10, S11, S12, S13: The authors need to provide the positions of their molecular weight standards in all the Western blots presented in this manuscript.
Done.

6) Figure 4A: “Methystat” should be changed to “Methylstat”.
Done.

Reviewer #3 (Remarks to the Author):

In this manuscript, Evdokiia and colleagues have investigated the role of the actin cytoskeleton in shaping the nuclear and nucleolar architecture during adipogenesis. They used primary human adipose stem cells and analyzed the nuclear and nucleolar remodeling process of adipogenesis. Although nucleolar structure appears to be correlated with translational activity during adipogenesis, additional experiments are required to determine whether cytoskeletal remodeling is essential for nucleolar architecture-mediated adipogenesis.

Here, we respectfully want to point out that the focus of our study is examine how nucleolar architecture is regulated in the context of adipogenic differentiation. We use adipogenic differentiation as a model system where cells undergo dramatic cytoskeleton reorganization, allowing us to assess the interplay between cytoskeleton and nucleolar remodeling in a physiological context. The objective of our study is not “to determine if cytoskeletal remodeling is essential for nuclear architecture-mediated adipogenesis”. We apologize if this was unclear to start with. Thus, we do not at any point claim that adipogenic differentiation is mediated by changes in nucleolar architecture.

Followings are specific comments to improve the study:

1. Even though the authors identified the nucleolar dynamics during adipogenesis and demonstrated a correlation between nucleolar changes and several time points of adipogenesis, they should carry out experiments that manipulate nucleolar dynamics to validate their effects on adipogenesis. Otherwise, it is unclear whether the changes in the nucleolus would be a cause or consequence of adipogenesis. Moreover, the biological significance of nucleolar number, size, or sub-compartmental organization during adipogenesis should be provided.

The biological significance of nucleolar number size and function in the adipose context has been previously described by us and others. We have shown that lncRNA *HOTAIR* knockdown in human ASC prevents actin cytoskeleton remodeling during early adipogenesis, and alters both translation rates and nucleoli morphology, resulting in a decrease in lipid storage in terminally differentiated adipocytes (Potolitsyna et al. Sci Rep 2022). Hence, our study together with the work of Audano et al. (JCB 2020) establish the biological importance of tightly controlled nucleolar function for the maintenance of adipocyte lipid storage capacity. To our knowledge, experimental set ups to manipulate nucleolar dynamics involve knockdowns of regulators of rRNA synthesis and/or components of the ribosome biogenesis machinery, which would imply the generation of multiple stable cell lines from our primary adipose stem cell system. Such manipulations would likely affect cell fitness and differentiation capacity, and would not allow to draw conclusions on a direct link between nucleolar architecture and differentiation efficiency. As mentioned above, the intent in this study is to use the adipogenic differentiation as a system where nucleoli are physiologically remodeled. We link this remodeling to both growth-arrest and actin cytoskeleton dynamics.

2. Considering the previous report that actin-associated genes are altered upon changes in lipid droplet locularity (Kim et al., Molecular and Cellular Biology 39.20, 2019: e00210-19), the authors should

investigate whether the enlarged and unilocular lipid droplet in adipocytes would affect cytoskeletal remodeling that results in nuclear and nucleolar changes.

While unilocular adipocytes cannot be obtained in a 2-dimensional differentiation system, *in vitro* differentiated adipocytes harbor lipid droplets of varying size. We now show significant correlations between maximal lipid droplet size and nuclear volume and shape (new Fig1. g,h,i), demonstrating that lipid droplet growth is a key driver of nuclear morphological remodeling in late adipogenesis.

3. The rationale to select day 3 for adipogenesis time point is unclear. Is it related to the periods of changes in the expression of adipogenic key regulators (PPARgamma, C/EBPalpha, etc.)?

Also, is there any link between key adipogenic transcription factors and the number or shape of nucleolus? The authors should properly address these issues.

The selection of day 3 as early adipogenic time point is based on (i) previous publications from our lab where we established that the adipogenic program is fully initiated by this time (Shah et al. BMC genomics 2014, Rønningen et al. Genome Res. 2015, Paulsen et al. Nat genet 2019, Madsen-Østerbye et al. Genome Biol 2022, Potolitsyna et al. Sci.Rep. 2022). This is supported by new transcriptomic data (see new Supplementary Fig. 1c); (ii) morphological changes, and the appearance of Perilipin-positive lipid droplets, allowing identification of differentiating cells by microscopy (see new Fig. 2a and Supplementary Fig. 4a). To the best of our knowledge, among key adipogenic transcription factors, only a minor translational isoform of CEBPA has been shown to localize in the nucleolus where it positively regulates rDNA expression (Muller et al. EMBO journal 2010), and no adipogenic transcription factor has been linked to changes in the number or shape of nucleoli.

4. The authors should provide appropriate data representing adipogenesis. For instance, they should examine lipid droplet formation via Perilipin1 and BODIPY staining (Figs. 1, 2, 4, Supplementary Figs. 1, 4) and measure the expression levels of adipogenic marker genes. These data are crucial to proving that the observed nucleolus changes would be associated with adipogenesis.

We thank the reviewer for the suggestion. To confirm the efficiency of our adipogenic differentiation protocol, we now provide a new figure (Supplementary Fig. 1) including (i) wide-field phase contrast images across the differentiation time course (Supplementary Fig. 1a) and Oil Red O staining of neutral lipids (Supplementary Fig. 1b) for both donors examined; (ii) a clustering analysis of differentially expressed genes over the adipogenic RNA-seq time-course showing an upregulation of genes pertaining to the Hallmark term “Adipogenesis” (Liberzon et al. Cell systems 2015) (Supplementary Fig. 1c) (iii) the gene expression profile of master adipogenic transcription factors (*PPARG*, *CEBPA*, *CEBPB* and *CEBPD*) over the differentiation time course (Supplementary Fig. 1d). Perilipin1 immunostainings in differentiating ASCs are presented in Fig. 2a and Supplementary Fig. 4a.

5. In Fig. 2, the authors should examine markers whether the cell cycle has been clearly arrested. For example, they should investigate the expression levels of Cyclin, Cdk, etc., under each experimental condition, referring to previous results analyzing cell cycle arrest in adipocyte differentiation (Reichert et al., Oncogene 18.2, 1999: 459-466).

Although mitotic clonal expansion is an important prerequisite to adipose differentiation in murine models of adipogenesis like 3T3-L1 cells, such mechanism is not relevant to human adipogenesis. The absence of proliferation after adipogenic induction has been documented in a recent publication from our lab (Madsen-Østerbye et al. Genome Biol. 2022), in agreement with our previous data demonstrating that exposure to cAMP-increasing agents inhibits proliferation of human adipose stem cells (Boquest et al. Mol Biol Cell 2008). In addition, we now present a clustering analysis of differentially expressed genes over the adipogenic RNA-seq time-course confirming the downregulation of cell cycle-related genes (GO:0071850 and GO:0000278) from D0 onwards (Supplementary Fig. 1c). We also confirm by flow cytometry analysis

of the DNA content the absence of mitotic cells at D0, and after 24h serum starvation (Supplementary Fig. 6b and 11).

6. The images of differentiating adipocytes mainly focus on the nucleus, making it difficult to recognize the cell's overall shape (Figs. 1a, 1e, Fig. 2, Supplementary Figs. 1a, 4a). Providing images taken at a lower magnification would be helpful.

We thank the reviewer for the suggestion and now provide larger field images for Fig. 1a. Larger field of view for Fig. 2 (now Fig. 3) were already presented as supplementary data (*now* supplementary figure S6). Figure 2 and 3 focus on nucleoli morphological remodeling, which is best appreciated at higher zoom level. However, Perilipin staining of lipid droplets confirms the adipose identity of cells at D3 and D15 time points. In addition, we now provide phase contrast images to show cell morphology for all differentiation time-points (see Supplementary Fig. 1a,b).

7. Proper explanation is required for different changes in nucleolar volume observed in Fig. 1f and Supplementary Fig. 3b. It seems that they would be the result from the same experimental design.

This study has been performed on primary adipose stem cells derived from two unrelated human donors. As shown in Fig. 4a and Supplementary Fig. 7a, while these two donors display the same trend in ribosomal protein gene expression during adipose differentiation, the fold increase in expression is more pronounced in Donor1 than Donor2. This result is even more striking when comparing mature adipocytes isolated from the patients' tissues (Supplementary Fig. 7b). This could be explained by rDNA copy number variation, which positively correlates with ribosomal gene expression in humans (Gibbons et al. Nat comm 2014). Similarly, rDNA copy number variation could potentially explain the differences in nucleolar volume between Donor1 and Donor2, both at the proliferative stage and in terminally differentiated adipocytes. Nevertheless, the changes in Nucleolus/Nucleus volume ratios during adipogenesis display identical trends for the two human donors, resulting in a single nucleolus occupying respectively $7,3\pm 3,2\%$ (Donor1) and $8,4\pm 2,1\%$ (Donor2) of the nucleus volume.

8. In Fig. 4, the drug treatment conditions are not described. Cellular morphology of the control group in Fig. 4 looks like proliferating adipose stem cells. To examine the effect of the cytoskeleton on nucleolar dynamics observed during adipogenesis, it is recommended that drugs be administered at day 0, when adipose stem cells are reached confluence and are growth-arrested.

The actin cytoskeleton is actively remodeled in the early stages of adipogenesis, and preventing or promoting this remodeling significantly impacts the capacity of adipose progenitors to undergo adipogenic differentiation (McBeath Dev Cell 2004, Chen et al. Stem cells research 2018, McColloch Sci Rep 2019). Thus, to examine the impact of cytoskeleton remodeling on nucleolar dynamics, we chose to disrupt or stabilize the actin cytoskeleton in proliferating ASCs, as now clarified in the figure legend. Indeed, applying these treatments during adipogenesis would not allow to discriminate a direct impact of cytoskeletal remodeling from any indirect effect linked to the enhancement or disruption of adipogenic differentiation. We show that actin cytoskeleton depolymerization in ASCs results in the formation of a single nucleolus, similar to what is observed in terminally differentiated adipocytes.

Updated figures

Figure 1

Figure 1. Nuclear and nucleolar remodeling during adipogenesis. **a** Immunofluorescence of lamin A/C, Phalloidin and DAPI stainings in differentiating ASCs (scale bar: 10 μm). **b,c,d**, Relative cytoskeleton volume, coherence, and average branch length measured from phalloidin signal ($***p < 0.0001$ vs D0, two-way ANOVA with Tukey's multiple comparison test; $n \geq 5$ fields of $2500 \mu\text{m}^2$). **e** Lamin A signals, normalized to γ Tubulin, quantified from Western blots ($*p < 0.05$, one-way ANOVA with Holm-Šídák's multiple comparisons; $n = 3$ experiments). **f** Nuclear volumes (voxel) measured from DAPI signal ($***p < 0.001$ vs D0, two-way ANOVA with Tukey's multiple comparison; $n \geq 30$ cells per time-point from 3 experiments). **e** ~~Immunofluorescence of~~ **g,h** Scatterplots of nucleus volume (voxel) or nuclear elongation vs maximal lipid droplet (LD) size (pixel), fit with linear regression. **i** Representative immunofluorescence images of Perilipin1 (PLIN1) and DAPI staining in D15 adipocytes (scale bar: 10 μm).

NCL PLIN1 DAPI

Figure 2

Figure 2. Nucleolar remodeling during adipogenesis

a Immunofluorescence of Nucleolin (NCL), Perilipin1 (PLIN1) and DAPI staining in differentiating ASCs (scale bar: 10 μm). **b** Scatter plot of nucleolar volume measured from Nucleolin immunostaining (** $p < 0.0001$ vs D0, ~~twoway~~two-way ANOVA with Tukey's multiple comparison; $n \geq 60$ cells per time-point from 3 experiments). **c** Scatter plot of the number of nucleoli per cell (~~twoway~~two-way ANOVA with Tukey's multiple comparison test; $n \geq 60$ cells per condition from 3 experiments). **d** Scatter plot of nucleolus-to-nucleus volume (No/Nu) ratio (** $p < 0.005$, *** $p < 0.0001$ vs D0, two-way ANOVA with Tukey's multiple comparison; $n \geq 60$ cells per condition from 3 experiments; ns, non-significant).

Figure 3

Figure 23. Cell cycle arrest triggers a rearrangement of nucleolar substructure.

a Immunofluorescence analysis of Nucleolin (NCL) and UBTF in proliferating (Pro) and D0 conditions (scale bar: 10 μm) ~~and **b** colocalization analysis.~~ **b** Line profiles from **a** (left panel) and average NCL fluorescence intensity at nucleoli border vs center (right panel) (**p < 0.01 two-tailed paired T test; n \geq 10 nucleoli per condition). **c** Immunofluorescence analysis of Nucleophosmin (NPM1) and RNA POL 1 (RPA194) in Pro and D0 conditions (scale bar: 10 μm) ~~and **d** colocalization analysis.~~ **e** ~~SRRF-Stream super-resolution.~~ **d** Line profiles from **c** (left panel) and average NCL fluorescence intensity at nucleoli border vs center (right panel) (**p < 0.01 two-tailed paired T test; n \geq 10 nucleoli per condition). **e** SRRF-Stream super-resolution microscopy images of Nucleolin and UBTF immunostainings in differentiating ASCs (scale bar: 10 μm). **f** Average number of UBTF foci per ~~cell~~ nuclei in differentiating ASCs, ~~quantified from (e) with Imaris software~~ (**p < 0.001 vs D0, one-way ANOVA with Tukey's multiple comparison; n \geq 3 fields per condition from two independent experiments). **g** Representative SRRF-Stream super-resolution microscopy images of Nucleolin and UBTF immunostainings in fasted and refed undifferentiated ASCs. ~~**h** Western blot analysis of NCL expression in fasted and refed conditions. γ Tubulin is shown as a loading control. **i** NCL signals normalized to γ Tubulin, quantified from western blots (non significant (n.s), Wilcoxon matched-pairs signed rank test; n = 3).~~

Figure 4

Figure 34. Translation-related genes expression and translation efficiency during adipogenesis. **a** Heatmap representation of relative gene expression (log₂ FPKM) for genes pertaining to the ribosome biogenesis (GO:0042254) and translation (GO:0006412) gene ontology pathways (differentially expressed genes across time, $p < 0.01$, eBayes method, limma package). Selected genes of interest are highlighted for each cluster. **b** SUNSET analysis of protein synthesis during adipogenesis with (+) or without (-) 1 μ M Puromycin. γ Tubulin is shown as a loading control. **c** SUNSET signals normalized to γ Tubulin, quantified from (b) (* $p < 0.05$, ** $p < 0.01$, two-way ANOVA with Tukey's multiple comparison test; $n = 3$). **d** SUNSET analysis of protein synthesis rates in fasted and refed conditions during adipogenesis. γ Tubulin is shown as a loading control. **e** SUNSET signals normalized to γ Tubulin, quantified from (d) (* $p < 0.05$, ** $p < 0.01$, two-way ANOVA with Sidak's multiple comparisons test; $n = 3$). **f** Western blot analysis of P-Thr389-S6K, total S6K and 4EBP1 in fasted and refed conditions during adipogenesis. γ Tubulin is shown as a loading control. **g** 4EBP1 signals normalized to γ Tubulin, quantified from western blots (*** $p < 0.001$, two-way ANOVA with Sidak's multiple comparisons test; $n = 3$).

Figure 5

Figure 45. Cytoskeleton structure defines the number of nucleoli during adipogenesis^{number}. **a** Phalloidin staining only (upper panel) or merged Nucleolin (NCL) immunostaining with DAPI (lower panel) in control condition, and after cytochalasin D (CytoD), Sorbitol or Methylistat treatments (scale bar: 10 µm). **b** Nucleus volume, **c** nucleoli number, **d** nucleoli volumes and **e** 0.5 µM or 5 µM, and Methylistat 1µM treatments in proliferating ASCs (scale bar: 10 µm). **b** Actin immunostaining only (upper panel) or merged Nucleolin (NCL) immunostaining with DAPI (lower panel) in control condition, and after Jaspakinolide 2µM treatment (scale bar: 10 µm). **c,d** Coherence measurements from **a** and **b**, respectively

(*** p < 0.001 vs Control, Two-way ANOVA with Tukey's multiple comparison test; n ≥ 20 cells from 3 independent experiments). e Nucleus volume (***) p < 0.001 vs Control, Two-way ANOVA with Tukey's multiple comparison test; n > 35 cells from 3 independent experiments) and f cell surface (pairwise t-test with holmberg adjustment for multiple testing; ≥ 10 000 cells from 3 independent experiments) in control condition, and after CytoD 0.5 μM or 5 μM, Methystat 1 μM and Japakinolide 2 μM treatments. g Nucleoli number, h nucleoli volumes and i nucleolus-to-nucleus (No/Nu) volume ratio measured from Nucleolin and DAPI in control condition, and after CytoD, ~~Sorbitol or Methylstat treatments~~ 0.5 μM or 5 μM, ~~Methylstat 1 μM or Jasplakinolide 2 μM treatments~~ (***) p < 0.001 vs Control, Two-way ANOVA with Tukey's multiple comparison test; n ≥ 35 cells from 3 independent experiments).

Reviewers' comments:

Reviewer #1 (Remarks to the Author):

I was already quite positive in my initial assessment of this work (Referee #1).

Globally the authors have addressed well most of the detailed comments of Referee #2 and Referee #3.

Some comments are not addressed directly, like showing more time points (Referee #2).

Nonetheless, there are sufficient novelty elements of quality in this work to make it a useful contribution.

Reviewer #2 (Remarks to the Author):

Overall, the authors have successfully addressed my comments and concerns. The only recommendation that I have is that the authors should provide some statistical analysis of the differences between the line scans reported in their new Figure 3. Other than that, I would say that this paper should be accepted for publication.

Reviewer #3 (Remarks to the Author):

The authors tried to address most comments and questions. However, they did not conduct new experiments to directly address several issues by arguing that additional experiments would be challenging and beyond the scope of the current study. Instead, they focused on clarifying their existing data and methodology to address the concerns raised by the reviewer.

1. Despite this, the revised study still lacks a robust biological meaning, primarily because they utilized adipocytes as a model for cytoskeleton reorganization rather than focusing on adipogenesis. They should expand the discussion in the main text to clarify the use of adipocytes as a model system for cytoskeleton reorganization, and emphasize the references (Potolitsyna et al., Sci Rep 2022; Audano et al., JCB 2020). In addition, additional experiments should be performed in other models of cytoskeleton reorganization (e.g., myocyte differentiation and embryonic stem cell differentiation) to strengthen the biological significance of this work.

2. The claim that disruption of the nucleolar structure during adipogenesis is attributable to a decrease in actin filament structure lacks clear evidence. Consequently, the assertiveness in the title and lines 172-175 appears to be aggressive; a more direct or conservative tone would be helpful not to exaggerate. Additionally, performing a parallel experiment with mature adipocytes, similar to the methodology in Figure 5, could provide insightful data.

Reviewers' comments:

Reviewer #1 (Remarks to the Author):

I was already quite positive in my initial assessment of this work (Referee #1). Globally the authors have addressed well most of the detailed comments of Referee #2 and Referee #3. Some comments are not addressed directly, like showing more time points (Referee #2). Nonetheless, there are sufficient novelty elements of quality in this work to make it a useful contribution.

We thank the reviewer for the positive feedback.

Reviewer #2 (Remarks to the Author):

Overall, the authors have successfully addressed my comments and concerns. The only recommendation that I have is that the authors should provide some statistical analysis of the differences between the line scans reported in their new Figure 3. Other than that, I would say that this paper should be accepted for publication.

Statistical analysis of Nucleolin and NPM1 distribution at nucleolus border vs center is presented as a boxplot for figure 3b and 3d (right panels). In addition, statistical comparison for each pixel along the line profile vs the center pixel within nucleoli has now been added on the line plots for Nucleolin and NPM1 (Fig 3b,d, left panels). Fluorescence intensities of Nucleolin and UBTF or NPM1 and RPA194 cannot be compared, as they result from immunological staining using different antibodies and imaging in distinct channels.

Reviewer #3 (Remarks to the Author):

The authors tried to address most comments and questions. However, they did not conduct new experiments to directly address several issues by arguing that additional experiments would be challenging and beyond the scope of the current study. Instead, they focused on clarifying their existing data and methodology to address the concerns raised by the reviewer.

1. Despite this, the revised study still lacks a robust biological meaning, primarily because they utilized adipocytes as a model for cytoskeleton reorganization rather than focusing on adipogenesis. They should expand the discussion in the main text to clarify the use of adipocytes as a model system for cytoskeleton reorganization, and emphasize the references (Pitolitsyna et al., Sci Rep 2022; Audano et al., JCB 2020). In addition, additional experiments should be performed in other models of cytoskeleton reorganization (e.g., myocyte differentiation and embryonic stem cell differentiation) to strengthen the biological significance of this work.

The discussion section has been expanded to highlight previous studies establishing the biological significance of actin cytoskeleton reorganization for adipogenesis, and the link between cytoskeleton reorganization and nucleolar function in the adipose context. We also want to point out that a characterization of nucleoli number and volume following osteogenic differentiation is presented in Supplementary Figure 5.

2. The claim that disruption of the nucleolar structure during adipogenesis is attributable to a decrease in actin filament structure lacks clear evidence. Consequently, the assertiveness in the title and lines 172-175 appears to be aggressive; a more direct or conservative tone would be helpful not to exaggerate. Additionally, performing a parallel experiment with mature adipocytes, similar to the methodology in Figure 5, could provide insightful data.

The title has been edited following the reviewer's suggestion and now more accurately reflects the results presented in the manuscript. The section title preceding lines 172-175 have been toned down, and an additional literature reference is provided to strengthen the notion of nucleoli mechanosensitivity.